



# The dynamics of marsh-channel slump blocks: an observational study using repeated drone imagery

Zhicheng Yang[1,2,3], Clark Alexander[1,2], Merryl Alber [1]

[1] Department of Marine Sciences, University of Georgia, Athens, GA 30602, USA
[2] Skidaway Institute of Oceanography, University of Georgia, Savannah, GA, 31411, USA
[3] Now at Division of Earth and Ocean Sciences, Nicholas School of the Environment, Duke University, Durham, NC, 27710, USA

*Correspondence to*: Zhicheng Yang (yang-thinkmore@outlook.com)

**Abstract.** Slump blocks are widely distributed features along marsh shorelines that can play an important role in marsh

dynamics. However, little is known about their spatial distribution patterns, nor their longevity and movement. We employed an Unmanned Aerial Vehicle (UAV) to track slump blocks in 11 monthly images (March 2020 – March 2021) of Dean Creek, a tidal creek surrounded by salt marsh located on Sapelo Island (GA, USA).  Slump blocks were observed along both convex and concave banks of the creek in all images, with sizes between 0.03 and 72.51 m$^2$. Although the majority of blocks were categorized as persistent, there were also new blocks in each image. Most blocks were lost through submergence, and

both decreased in area and moved towards the center of the channel over time. However, some blocks reconnected to the marsh platform, which has not been previously observed. These blocks were initially larger and located closer to the marsh edge than those that submerged, and increased in area over time. Only 13 out of a cohort of 61 newly created blocks observed in May 2020 remained after 5 months, suggesting that most blocks persist for only a short time. When taken together, the total area of new slump blocks was 886.13 m$^2$ and that of reconnected blocks was 652.45 m$^2$. This resulted in a

net expansion of the channel by 233.68 m$^2$ over the study period, accounting for about 66% of the overall increase in the channel area of Dean Creek, and suggests that slump block processes play an important role in tidal creek channel widening. This study illustrates the power of repeated UAV surveys to monitor short-term geomorphological processes, such as slump block formation and loss, to provide new insights into marsh eco-geomorphological processes.

# 1 Introduction

Salt marshes are globally valuable ecosystems, serving as crucial interfaces between marine and upland environments (Murray et al., 2022). They are important in nutrient cycling, shore protection, and carbon sequestration, and they also provide nursery habitat for commercially important fish and shellfish (e.g., Barbier et al., 2011; Chmura et al., 2003; Kirwan and Mudd, 2012; Möller et al., 2014). Marshes are dynamic environments and the area of vegetated marsh can change over

time, not only as the result of progradation or retreat of the open-fetch marsh edge (e.g., Marani et al., 2011; Mariotti and Fagherazzi, 2013; Schwimmer, 2001; Tommasini et al., 2019; Yang et al., 2022), but also due to widening and contracting of interior channels (Burns et al., 2021a; Chen et al., 2011; D'Alpaos et al., 2005; Zhao et al., 2022). These changes in marsh-edge geomorphology can influence drainage patterns (D'Alpaos et al., 2005; Stefanon et al., 2012, 2010; Zhou et al., 2014)





and the marsh sediment budget (Kirwan and Guntenspergen, 2010; Mariotti and Carr, 2014; Yang et al., 2023), with broad

implications for habitat provisioning, carbon storage, and other ecosystem services.

Slump blocks (see the left middle panel in Fig. 1) are vegetated sedimentary units that have broken from the marsh platform and occur as small islands in the adjacent channel. Slump blocks have been observed along marsh shorelines in many areas and can be quite common, and have been described, for instance, in New England (Houttuijn Bloemendaal et al., 2021; Redfield, 1972), San Francisco (Fagherazzi et al., 2004; Gabet, 1998) and the Netherlands (Koppel et al., 2005). Much

attention has been focused on blocks formed along open fetch marshes that are exposed to wave attack (Allen, 2000; Bendoni et al., 2016; Francalanci et al., 2013; Koppel et al., 2005; Zhao et al., 2017), but blocks are also seen in low-energy environments (FitzGerald and Hughes, 2019; Houttuijn Bloemendaal et al., 2021; Li and Pennings, 2016). In Georgia marshes, investigators documented blocks that separate from the marsh platform and then creep down the bank at an average rate of 16 cm/month (Frey and Basan 1978, Letzsch and Frey 1980b). Slump block formation can result from a variety of

factors, such as bank undercutting, variations in water level, seepage erosion and animal activity (e.g., Francalanci et al., 2013; Frey and Basan, 1978; Gasparotto et al., 2022; Gong et al., 2018; Kirwan and Murray, 2007; Rinaldi and Casagli, 1999; Zhao et al., 2022, 2021). Slump blocks can also be formed through freezing and thawing (Argow et al., 2011), and Deegan et al. (2012) reported vegetated blocks that broke from the marsh platform as the result of nitrogen fertilization, which decreased bank stability as the result of increased above-ground biomass and reduced below-ground biomass of the

vegetation colonizing channel banks.

Slump blocks can play an active role in marsh dynamics, potentially contributing to channel erosion and the lateral retreat of marsh boundaries (Deegan et al., 2012; Frey and Basan, 1978; Kirwan and Murray, 2007; Zhao et al., 2022). The fate of the material comprising slump blocks is largely unknown, but it may be incorporated into the sediments of the creek, redeposited on the marsh platform, or exported out of the system (Mariotti and Carr, 2014; Yang et al., 2023). Creekbank slumping also

disturbs marsh habitat: Li and Pennings (2016) found that slumping affected about 16% of long-term vegetation monitoring plots in Georgia marshes, with what they termed either an "initial" (crevices observed on the marsh edge) or  "terminal" (plot collapsed into the creek) slump. The formation of a slump block has implications not only for the survival of the vegetation on the block (which may eventually drown), but also the associated marsh organisms. Cracks produced by bank slumping might also provide a temporary refuge or a barrier for fish and other nekton (Nelson et al., 2019).

Despite their potential importance, little is known about slump block distribution over space and time, nor is there much information on their longevity or movement. This stems in part from the fact that slump blocks can be difficult to access in the field and so most previous work has been limited in scope. In this paper, we took advantage of unmanned aerial vehicles (UAVs), which offer a cost-effective way to conduct synoptic observations of salt marshes (e.g., Dai et al., 2021; Doughty et al., 2021; Lynn et al., 2023; Pinton et al., 2020). We used 11 high-resolution UAV images taken over a one-year period to

track slump blocks along a tidal creek in a Georgia salt marsh. We had three main objectives: 1) to describe the spatial and temporal distributions of slump blocks; 2) to characterize the "life cycle" of slump blocks by tracking newly formed blocks over time; and 3) to assess whether there is a relationship between slump block formation and changes in channel area. Our





results offer new insight into the dynamics of these blocks, and suggest that they are important components of the ecological and geomorphological processes of salt marshes.

## 2 Methods

### 2.1 Study site

This analysis was carried out on Sapelo Island, Georgia, a barrier island in the southeastern USA (see Fig. 1). The average rate of sea level rise (SLR) is about 3.5 mm/year, based on records measured at the NOAA Fort Pulaski tidal gauge from 1935 to 2023 (National Oceanic and Atmospheric Administration, 2023). Dean Creek, the focus of this study,  is a salt marsh tidal creek on the south end of Sapelo Island that is constrained by the Pleistocene upland to the west and Holocene recurved sand spits to the east. Dean Creek is about 40 m wide at its mouth where it connects to Doboy Sound.  Water levels in this area are predominantly controlled by semi-diurnal tides with an average tidal range of approximately 2.5 m. Tidal flow is ebb-dominant, and there are several ebb-oriented point bars located in areas where the ebbing flow deposits sediment downstream of convex shorelines (Fig. 1). These point bars are typically sandy, and exhibit sand waves demonstrating their non-cohesive nature. Finer-grained sediment accumulates along the creekbank in the lee of these point bars. Sandier sediments are also deposited downstream of tight-radius changes in channel direction, where large scale bedforms are also observed (Letzsch and Frey, 1980a).

Dean Creek is surrounded by salt marsh habitat, which is dominated by the cord grass *Spartina alterniflora* ('*Spartina*' hereafter), with tall-form plants (>70 cm) growing primarily along creekbanks and medium- (30-70 cm) and short-form plants (<30 cm) found on the main marsh platform. The slump blocks, which detach from the creekbank, are colonized by tall-form plants. For this study we focused on an 850 m segment along the length of Dean Creek, as part of an ongoing research project to understand salt marsh disturbances within the Georgia Coastal Ecosystems Long-Term Ecological Research (GCE-LTER) Program.

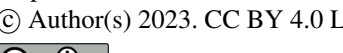

**Figure 1: Overview of the research area. The inset on the top left shows the southeastern coast of the USA and the location of Dean Creek (yellow star) on Sapelo Island. The main map shows Dean Creek and surrounding marshes captured in July 2020 using a DJI Matrice 210 UAV with a MicaSense Altum (Near Infrared, central wavelength = 840 nm) and the positions of slump blocks digitized from each image of the study. The yellow triangles indicate the positions of point bars and the yellow arrow indicates banks with a wide distribution of oysters. The inset on the middle left shows an example of slump blocks (photograph by Merryl**



**Alber). The inset on the bottom right shows the numbered reaches along the centerline of Dean Creek, separated by points where the curvature equals zero.**

## 2.2 Image analysis

A series of 11 UAV images of Dean Creek were used for this analysis. Images were flown monthly between March 2020 and March 2021, with the exception of April and November 2020, when the UAV was out of service (Table S1). Images were acquired using a DJI Matrice 210 UAV equipped with a MicaSense Altum sensor, as described in detail in Lynn et al. (2023). Briefly, all images were acquired during morning low tides within 1-2 hours of solar noon. Tides at the time of the flights averaged 1.7 m below MSL, with a minimum of 1.2 m below MSL. Permanent ground control points were installed

for this effort, and these were used to produce georeferenced images using Pix4D software. Pixel resolution was 0.05 m, and georectification resulted in an average root mean square error of $0.0025 – 0.0050 \text{ m}^2$.

The intact marsh boundary and the perimeters of slump blocks in each image were manually digitizedusing ArcGIS 10.8. The intact marsh boundary was defined based on areas of continuous vegetation along the edge of the creek without any bare gaps or crevasses wider than 0.3 m. Slump blocks were defined as vegetated units surrounded by water. We set a minimum

block size of $0.0025 \text{ m}^2$ so that each block contained at least 1 pixel. We also set a 0.3-m minimum distance from the nearest intact marsh edge to delineate a disconnected block. These metrics were chosen based on what could be readily distinguished as distinct from the bank in the imagery as well as preliminary field measurements that demonstrated that incipient blocks were at least that far from the channel edge before functioning as a separate unit. We assessed the accuracy of our estimate of slump block size by randomly selecting 6 blocks of different sizes and manually digitizing their boundaries 10 times. The

coefficient of variation of these estimates ranged from 2.5 to 9.6% and averaged 4.9% (see Table S2 in Supporting Information), which provides a measure of the error associated with the manual estimation of slump-block size.

The first UAV image (acquired in May 2020) served as the baseline image for channel segmentation into reaches and further analyses of shoreline change. Specifically, the digitized intact marsh edge was used to convert the image into a binary map (water or marsh) using ArcGIS 10.8. Next, the centerline of the channel, which represents the main axis of Dean Creek, was

generated by applying the skeletonization procedure to the channelized area in Matlab R2020a software (Kerschnitzki et al., 2013). The creek was then segmented into 12 reaches at the location of inflection points of the channel centerline where the curvature of the centerline equaled 0. The curvature at each pixel of the channel centerline was estimated by using the method proposed by Marani et al (2002). The centerline length of these 12 reaches ranged from 40 to 142 m, and averaged 72 m. Banks were designated as concave or convex within each reach based on their planar morphology.

## 2.3 Spatial and temporal distribution of slump blocks

The digitized imagery provided information on the location, size, and number of slump blocks observed in the study area in each image. We used these data to characterize blocks in terms of their size frequency distribution and evaluate how the





number and cumulative area varied over time. We also used the location information to quantify the number and cumulative area of blocks in each reach of the channel. Finally, we evaluated whether bank shape (convex or concave) influenced the

spatial distribution of blocks by comparing the number and cumulative area of blocks along each bank, normalized to the length of the nearest bank segment within each reach.

## 2.4 Tracking slump blocks

We followed individual slump blocks from image to image in order to understand how they changed over time in each observation interval. We classified blocks into four categories: "new", "persistent", "submerged" and "reconnected". New

blocks were those that were only present in the latter image of an observation interval and thus were newly formed (Fig. 2 a, b). Note that new slump blocks could not be identified in the initial image (March 2020) since we did not have an antecedent image for comparison. Persistent blocks were those that were present in both images (Fig. 2 e, f). Submerged blocks were those that were no longer present in the latter image (Fig. 2 g, h). Reconnected blocks were those that were no longer separated from the marsh in the latter image (Fig. 2 j, k). This latter category was unexpected, as we had assumed that blocks

would all move down the bank towards the base of the channel over time and eventually be submerged. We therefore visited several of the reconnected blocks using Real-Time Kinematic GPS positioning in June 2023, more than two years after the end-point of the study, to confirm that these blocks had in fact been reincorporated into the intact marsh (Supplementary Fig. S1).

We used the classifications to track blocks in each category so that we could determine the cumulative area of blocks that

were newly formed, persisted, submerged or reconnected between images. The size of blocks can either increase (Fig. 2 e, f) or decrease (Fig. 2 f, g) over time, so changes in the cumulative block area could be positive or negative. In order to calculate the net change in the area of blocks from the previous image, we summed the changes in persistent blocks, as well as the area of blocks that had submerged within the interval. We also observed both the merging and splitting of blocks between images (Fig. S2 in Supporting Information), which affected the number but not necessarily the cumulative area of

the blocks. We therefore rely mainly on areal changes to interpret our results.

We compared the initial characteristics of blocks with different destinies (i.e., submerged and reconnected with the marsh platform) by selecting those blocks that were identified as newly produced and either submerged or reconnected within the observation period. We measured their distances to the intact marsh edge at the time they were first observed, which we considered their initial gap width, as well as their initial size, to compare the properties of different block types.

Finally, we followed a "cohort" of new slump blocks (hereafter 'cohort blocks'), identified in our first repeat aerial survey in May 2020. This allowed us to estimate how long a new slump block is likely to last and to characterize the change in the cumulative area of cohort blocks over time. We also used cohort blocks with a lifespan longer than one month (i.e., occurred in at least two consequent images) to analyze the rates of movement and changes in size and location of submerged and reconnected blocks. To analyze changes in their locations, we tracked them in each image by measuring their shortest

distance to the fixed intact marsh boundary, which was digitized based on May 2020 image, and then used this information





to calculate the rate of movement towards the channel centerline of each block type over time using linear regression in Matlab R2022a. The rate of change in area over time for each block type was also analyzed by linear regression, again based on Matlab R2022a software.

**Figure 2: Examples of changes in slump blocks over time.** A denotes digitized area of the blocks in each panel; yellow lines represent slump block boundaries and black lines represent the edge of the intact marsh. Panels (a-d) depict ormation of a new slump block, which was first observed on 3/18/2020 (panel b) and was still present on 3/24/21 (panel d). The location of the block before it broke from the marsh is denoted by a dotted yellow line in panel a. Note the change in the shape of marsh edge once the block formed. Panels (e-h) depict changes in the location and area of a slump block as it submerged. The dashed yellow line in panel h represents the boundaries of the location of the block in the previous observation., Panels (i-l) depict reconnection of a slump block, resulting in a lateral extension of the marsh edge. The dashed yellow line in panel k represents the boundaries of the location of the block in the previous observation. The area of the block in panel j was used to represent the area of the block reconnected with the marsh platform on the next observation date (e.g., 06/17/2020). Scale bars in panels a, e, and i apply to the top, middle, and bottom rows, respectively.



### 2.5 Slump block contribution to changes in channel area

We estimated the change in channel area over the course of the study by comparing the channel edge location in the first
(March 2020) and final (March 2021) images. Although in most places the channel had widened over time, there were a few
places where it had narrowed. We then calculated the area that had been removed from the creekbank in the form of slump
blocks by summing the area of newly produced blocks (regardless of whether they were persistent or submerged) and
subtracting the area of blocks that were reconnected. We compared these values to estimate what percentage of the change in
channel width could be accounted for by the process of slump block formation. We did this calculation for each reach of the
channel as well as for the entire study area as a whole.

### 3 Results

### 3.1 Spatial and temporal distribution of slump blocks

Slump blocks were present in each of the 11 images of Dean Creek that we analyzed for this study (see Fig. 1 with the UAV
image acquired in July 2020 and Fig. S3 in Supporting Information). Blocks exhibited a log-normal size distribution (Fig.
S4). They ranged from $0.03 - 72.51$ m$^2$, with an average size of 5.08 m$^2$. There were an average of $182 \pm 30$ blocks per
image (Fig. 3a), with an average cumulative area of $879.74 \pm 118.55$ m$^2$ (Fig. 3b). The number of blocks fluctuated over the
course of the study year, with a maximum in June and July 2020 (223 blocks) and a minimum in December 2020 (146
blocks, see Fig. 3a), whereas the cumulative area ranged from a maximum of 1131.92 m$^2$ in June 2020 m in June to a
minimum of 757.24 m$^2$ in February 2021 (Fig. 3b). Given that this is only one year of observations, it is too early to ascribe a
seasonal pattern to these results.



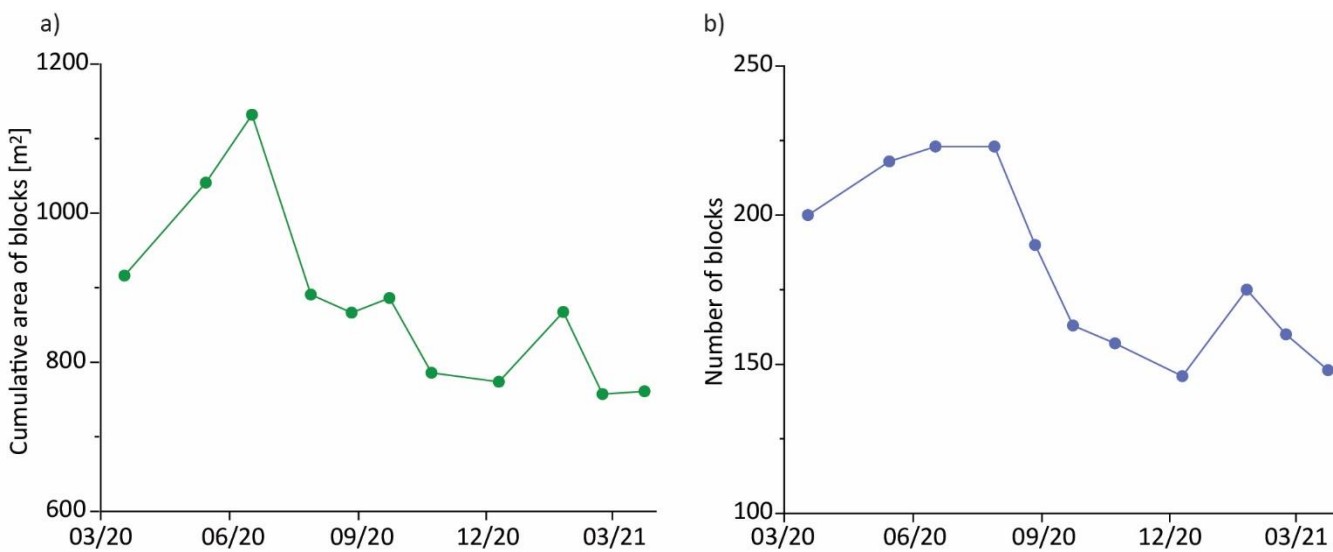

**Figure 3: Temporal changes in (a) cumulative area and (b) number of slump blocks digitized in each image over the course of the study. X-axes represent the acquisition date of each image with a format of mm/yy.**


Blocks were observed along both banks of the creek, with almost all reaches affected over the course of the study (Fig. 1 and Fig. S3). However, there was considerable spatial and temporal variation in their distribution, with the largest cumulative block area (Fig. 4a) and most blocks (Fig. 4b) in the middle reaches (3-7) and another peak in reach 11, which was further upstream. Interestingly, there were very few blocks in areas affected by ebb-oriented point bars (Figs. 1 and 4). The most

striking example is on the eastern bank along reach 10 where large-scale sand waves downstream of an ebb point bar are evident. No slump blocks were observed along this bank over the entire course of the study. There were also fewer blocks in reaches 1-2, which are closest to the mouth (Figs. 1 and 4). Although we predicted that more blocks would be associated with concave as compared to convex banks due to the potential undercutting of the creekbank, we found no evidence of such (Fig. S5 in Supporting Information).




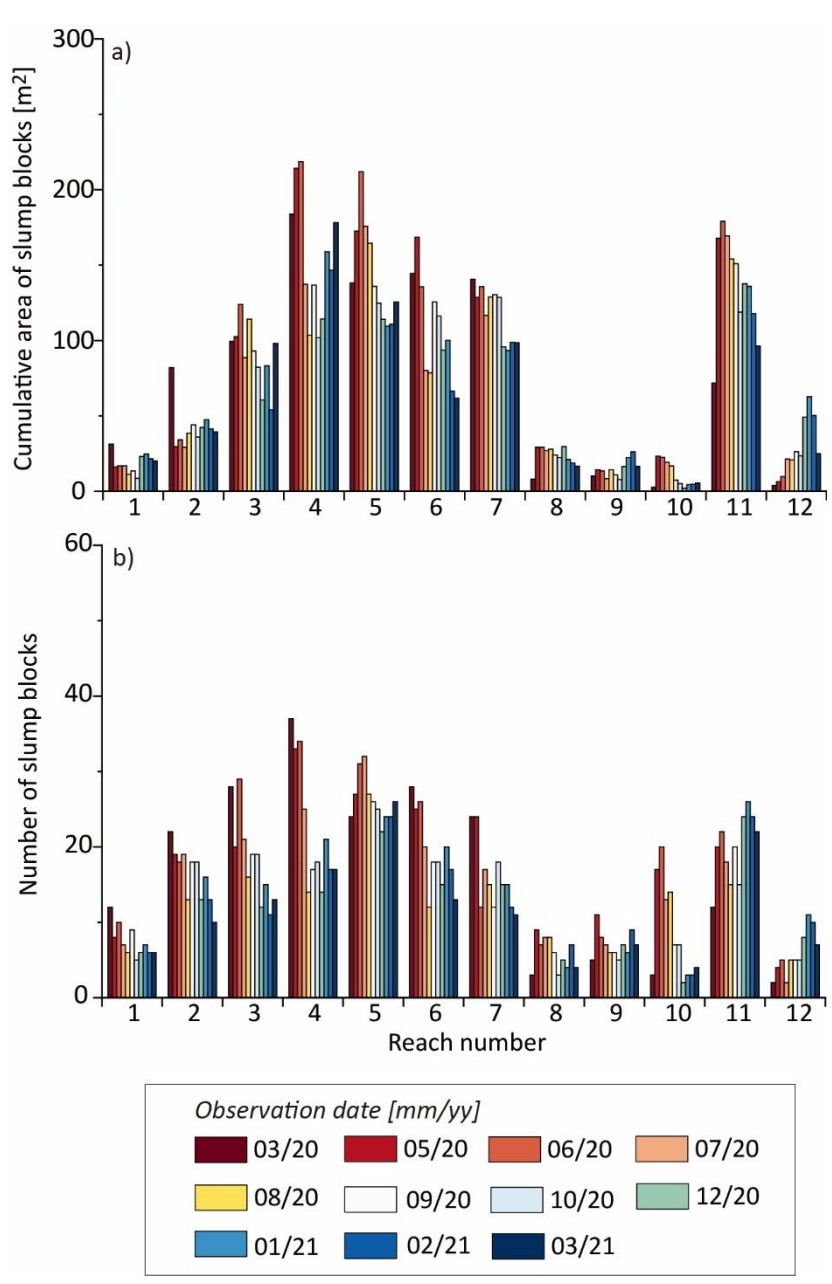

**Figure 4: Spatial and temporal distribution of (a) cumulative area and (b) number of slump blocks digitized over the course of the study, separated by reach (see inset, Figure 1).**





## 3.2 Tracking slump blocks

Classifying slump blocks by category allowed us to assess blocks that persisted from image to image, those that were new, and those that were lost, either through submergence or reconnection. The majority of blocks in each image were classified as persistent. There were an average of $141 \pm 31$ persistent slump blocks in each image (= 1410 total, see Fig. 5a), with an average cumulative area of $787.42 \pm 137.07$ m$^2$ (Fig. 5b). There were a total of 270 new blocks observed over the course of the study, with the highest number of new blocks appearing in May 2020, September 2020, and December 2020 images (Fig. 5c). The highest cumulative area of new blocks was observed in the December 2020 image, followed by those first observed in May 2020 (Fig. 5d). An average of $27 \pm 18$ new blocks (Fig. 5c) appeared in each image, with an average cumulative area of $88.61 \pm 69.27$ m$^2$ (Fig. 5d). A total of 322 blocks were lost to submergence (Fig. 5e), with the highest losses appearing in two intervals, i.e., from July to August 2020 (60 lost) and from October to December 2020 (57 lost).

As described in the methods, some blocks increased in area while others decreased (Fig. 2). The overall result (Fig. 5f) was that the net change in cumulative slump block area was positive from March to June 2020 and again from February to March 2021, despite the submergence of blocks within each of these intervals (Fig. 5e). When summed over the entire study period, there was a cumulative loss of 407.91 m$^2$ of block area (as the result of the difference between positive and negative values, Fig. 5f). In addition, a total of 107 blocks (Fig. 5g) with a cumulative area of 652.45 m$^2$ (Fig. 5h) were reconnected over the course of the study. If we compare blocks lost via submergence versus through reconnection to the marsh platform, we find that, although there are a greater number of blocks lost to submergence, given their generally small size, the overall loss in block area is primarily driven by the reconnection process.







**Figure 5: Changes in number (left column) and area (right column) of blocks with different fates over the course of the study. Panels a, c, e, and g show the number of persistent, new, submerged, and reconnected blocks that were identified during each observation interval. Panels b, d, and h show the cumulative area of persistent, new, and reconnected blocks that were identified during each observation interval. Panel f shows the net change in block area calculated by subtracting the area of submerged and reconnected blocks from the cumulative area of blocks in the previous image. X-axes represent the time interval of image acquisitions with a format of mm/yy – mm/yy.**

The comparison of the initial properties of blocks that submerged (Fig. 6a) with those that were lost through reconnection (Fig. 6b) revealed differences in slump blocks with these different fates. First, the initial size of blocks that were lost to submergence ($1.45 \pm 1.63$ m$^2$) was significantly smaller ($p < 0.05$, one-way ANOVA) than that of those that eventually reconnected ($5.49 \pm 4.53$ m$^2$) (Fig. 6c). Second, blocks that were lost to submergence were significantly further ($p < 0.05$) from the marsh edge ($1.34 \pm 1.25$ m) when first observed compared to those that eventually reconnected ($0.49 \pm 0.21$ m, see Fig. 6d).









**Figure 6: Comparison of blocks with different destinies. (a-b) are maps of blocks that finished their lifespan (i.e., from newly produced to completely submerged or reconnected with marsh) within the study period; (c-d) Box-whisper plots of initial size and initial gap width of submerged and reconnected blocks. Data are presented with the average value, quartiles and extremes. The p-value in each panel is the result of a one-way ANOVA test conducted to test the hypothesis of no difference between the means of the two groups.**


The cohort of new blocks that we tracked starting in May 2020 provided information on the rate of disappearance of blocks over time (Fig. 7). During the first 5 months (from May to October) there was a linear reduction in the number of blocks ($R^2$ = 0.97), from 61 to 13. Most of these (44) had submerged while 4 had reconnected (Fig. 7a). During the first 5 months, the cumulative area of blocks decreased linearly ($R^2$ = 0.97) from 184.52 to 109.85 m$^2$ (Fig. 7b). The cumulative area of the

submerged blocks over this period was 49.89 m$^2$, whereas that of the reconnected blocks was 27.58 m$^2$. In both cases the rates of change decrease after 5 months, with the remaining blocks largely persisting. At the end of the study period (after 10 months), 10 blocks persisted, with a cumulative area of 75.67 m$^2$, which accounts for 16% by number and 41% by area of the original cohort.





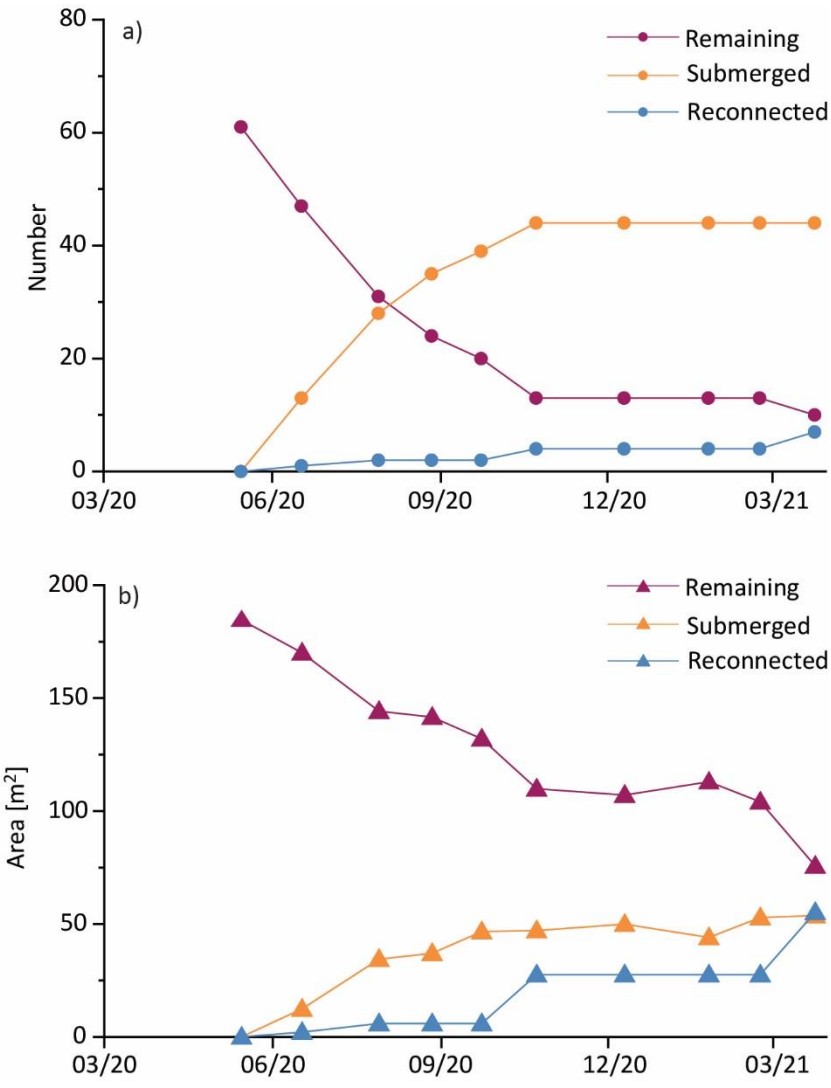


**Figure 7: Temporal changes in the number (a) and cumulative area (b) of a cohort of new blocks first observed in May 2020. Remaining blocks were present in the creek through the last observation (April 2022); submerged were blocks that were no longer present in the creek; reconnected were those that merged with the marsh platform. X axes represent the date of image acquisition with the format of mm/yy.**


When we followed cohort blocks with different fates we found that the submerged blocks decreased in area over time (Fig. 8a) whereas those that eventually reconnected increased in area (Fig. 8b). In addition, most of the blocks that were submerged moved towards the channel center (Fig. 8c), whereas those that reconnected were likely to move toward the bank (Fig. 8d). Interestingly, the rate of movement of the submerged blocks towards the channel center (Fig. 8c) was an order of

magnitude faster than the rate of movement of the marsh-facing edge of reconnecting blocks toward the intact marsh platform (Fig. 8d).







**Figure 8: Frequency distributions of changes in area (a, b) and rate of movement (c, d) of a cohort of new blocks first observed in**
**280** **May 2020. Blocks that submerged over the course of the study decreased in size (a) and tended to move towards the center of the**
**channel (c) whereas those that reconnected increased in size (b) and tended to move towards the bank (d).**

## 3.3 Slump block contribution to changes in channel area

Overall, we estimate that the channel area increased by 354.98 m$^2$ between May 2020 and May 2021, which follows the
**285** same trend as the long-term increase estimated from a series of aerial photographs collected between 2013 and 2018 (Fig. S6

in Supporting Information). Although this translates to an average increase in channel width of 0.41 m, the change in channel area varied by reach (Fig. 9): reach 6 showed a decrease of 90.57 m$^2$ in channel area over the course of the study whereas reach 10 showed an increase of 128.98 m$^2$. There was a positive linear relationship ($R^2 = 0.55$ and $p < 0.01$) between the net

area of marsh lost due to the formation of slump blocks and the observed increase in channel area within each reach, with net area calculated as the total area of newly produced blocks minus the area of reconnected blocks measured over the study period. Note that the net change due to slump blocks was negative in reaches 2, 6 and 7 as a result of block reconnection with the marsh platform, which matched the decrease in channel width in the associated reaches. When taken together, the total area of new slump blocks was 886.13 m$^2$ and that of reconnected blocks was 652.45 m$^2$, resulting in a net loss due to slump blocks of 233.68 m$^2$. This represents 66% of the overall increase in channel area, and suggests that the slump block

generation process plays an important role in tidal creek channel widening.

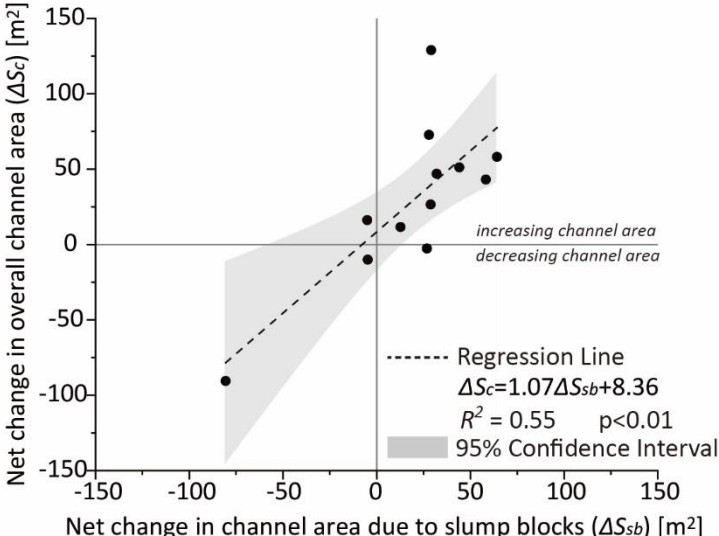

**Figure 9: The relationship between the net change in channel area due to slump-blocks processes (the difference between new the block area and area of blocks reconnected with the marsh platform, $\Delta S_{sb}$) and the observed change in channel area ($\Delta S_c$) in**
**different channel reaches (see Figure 1). The lines through zero separate areas that showed loss (negative values) and gain (positive values) of channel area. The linear regression is also included along with the 95% confidence interval (represented by the light grey area).**

## 4 Discussion

This analysis yielded several important insights regarding slump blocks in salt marshes. First, slump blocks can be ubiquitous along tidal creeks. Second, although most slump blocks submerge over time, some of them reconnect to the intact



marsh platform. Third, these slump blocks are dynamic and generally persist for less than 6 months. Finally, the formation and loss of slump blocks can be an important contributor to creek widening. We examine each of these findings below.

### 4.1 Spatiotemporal distribution of slump blocks

The presence of vegetated slump blocks in salt marshes has been described in publications dating back to the 1970s (Redfield, 1972), and there have been studies over the years on this and related phenomena (e.g., Bendoni et al., 2016; Francalanci et al., 2013; Gabet, 1998; Gao et al., 2022; Letzsch and Frey, 1980b; Zhao et al., 2022). However, improvements in UAV and computing technology coupled with repeat observations have now provided us with enhanced capabilities to characterize the spatial and temporal dynamics of these blocks. Here we report that vegetated slump blocks were present

along all reaches of Dean Creek on Sapelo Island in all 11 images encompassing the one-year time frame of the study. However, there were more blocks with greater cumulative area in reaches 3-7 and 11 than the other reaches, and most reaches had their highest number and area of blocks in the first 3 images (March 2020 - June 2020).

Although this was an observational study, we can use the patterns of slump block occurrence (and non-occurrence) to explore the factors that may contribute to slump block formation. One interesting pattern in this regard was that few to no

blocks were found downstream of, and in the lee of, ebb point bars (Fig. 1 and Fig. S3). We suggest that the ebb-oriented point bars serve to protect the adjacent bank from erosion, provide a more quiescent environment in which finer-grained sediment can accumulate, and create a gentle slope that *Spartina* can more easily colonize, as compared to the rest of the creek. Although limited in number, most of the blocks that were formed in these areas ended up reconnecting, and reach 6 in particular, where a large number of blocks reconnected, was the one reach where the channel area decreased over the course

of the study. We also saw fewer blocks in reaches 1 and 2 at the mouth. This is an area with abundant oyster reefs (Fig. 1 and Fig. S3), that may protect and stabilize the bank to prevent blocks from forming.

We expected that undercutting of the creekbank edge by currents would be important in the formation of slump blocks. This has been observed in other systems (Francalanci et al., 2013; Schwimmer, 2001), and rotational failure as a consequence of undercutting was suggested as a possible cause of bank slumping of Georgia marshes (Frey and Basan, 1978). However, we

did not find a higher density of slump blocks associated with concave as compared to convex banks where we would expect the currents to be faster (Fig. S5). This suggests that bank undercutting is probably not the primary reason for bank slumping in this system, and that other processes are likely involved. In fact, large-scale pore water circulation toward the creekbank has been documented throughout other similar Georgia marshes (Jahnke et al., 2003), which may have the potential to reduce bank stability and promote the formation of slump blocks (Mariotti et al., 2019; Zhao et al., 2022). However, the

primary reason for the bank slumping in this area is still unclear and deserves future analyses.

Although this clearly needs further study, our field observations show numerous cracks (see Fig. S7) along the marsh edge that make the bank unstable, and we suggest that the slump block formation that we observed is likely a function of the local slope in combination with the load exerted by the tall, aboveground vegetation and saturated, unconsolidated muddy sediments.





## 4.2 Tracking slump blocks

As far as we know, this is the first demonstration that slump blocks can reconnect to the creekbank (but see Redfield, 1972, who speculated that this might occur). This fate was unexpected, as we assumed that all slump blocks would be submerged over time. We were initially concerned that this could be an artifact if the vegetation had leaned over and obscured the gap in the UAV imagery, especially during the growing season. However, we confirmed this finding through both field observations and additional UAV imagery in June 2023, more than two years after the end of the study. In March 2020, the gap between the bank and the slump block displayed in Figs. 2i-l was approximately 80 cm (measured in ArcGIS), whereas in June 2023 the creekbank was continuous in this location, with no evidence of a block. Moreover, the maximum distance between vegetation patches was approximately 3 cm (see Fig. S1g).

The ability to track slump blocks from image to image provides information on the dynamics of block persistence and fate. The number and cumulative area of blocks varied, with an average of 27 new blocks, and 141 persistent blocks in each image. Blocks were also lost during each interval, either to submergence (an average of 32 blocks) or to reconnection (an average of 11 blocks). The picture that emerges is quite dynamic, particularly as some blocks increased in area whereas others lost area between images, not to mention splitting and/or merging. Although we found an overall net loss of block area over the course of the year, there were several intervals where blocks that were tracked from the previous image actually had a net increase in area even when the loss of submerged blocks was accounted for (note that for this analysis reconnected blocks were not considered lost from the system). These observations show that slump blocks are not stable and can follow different trajectories over time, which would not have been captured with a one-time survey.

We cannot estimate an average lifespan for the blocks as 10 of the cohort blocks that we started tracking in May 2020 remained at the end of the study. However, most were lost during the first seven months: only 13 of the original 61 remained in December 2020, and their cumulative area had decreased from 184 to 107 m$^2$, after which time the loss rate decreased. This lifetime is longer than that for slump blocks observed on unvegetated intertidal areas, where blocks tend to reach the channel base immediately (Gao et al., 2022). The longer persistence in the marsh is likely due to the presence of vegetation, as it has been shown that the associated root and rhizomes may serve to counterbalance gravity and a large portion of the shear stress exerted by water flow, thereby preventing the blocks from detaching too quickly or being eroded away immediately (Brooks et al., 2021; Chen et al., 2012; Simon et al., 2006). It should be noted, however, that blocks found in Plum Island, MA, have been tracked for multiple years (Deegan et al., 2012; Mariotti et al., 2019). This highlights the fact that slump blocks in different systems are not necessarily the same in physical character or behavior.

Finally, the cohort study provided a way to compare blocks of a known fate (submergence or reconnection) that were tracked over their entire life cycle. Although most of these were submerged (=44) as compared to reconnected (=7), the initial size of the blocks that eventually submerged was smaller than those blocks that finally reconnected with the marsh (average 1.45 vs. 5.49 m$^2$). This demonstrates that there were many small blocks that eventually submerged and a few larger ones that reconnected. The fact that larger blocks were more likely to reattach might be related to the larger area of *Spartina*, which




may have increased the strength of their initial attachment to the marsh platform and slowed the movement toward the channel base, thus enhancing the opportunity for sediment to accumulate within the gap.

When first observed, those that submerged were further from the creekbank (average distance = 1.34 m) than those that reconnected (average distance = 0.49 m). There were also differences over time: Blocks that submerged slowly lost area and generally moved towards the center of the channel, whereas those that reconnected gained area and exhibited a marsh-facing boundary that advanced toward the creekbank, as vegetation filled the gap between the block and the intact marsh platform. We suspect that the initial gap width is positively related to the slope of the bank, i.e., blocks that submerge are more likely

to be found in areas with steeper slopes than those that reconnect.

The fact that both the growth direction and rate of movement of reconnecting blocks were different than that of submerging blocks suggests that these blocks are affected by different processes. Submergence is likely due to a decrease in elevation as the block moves downslope. When the block is low enough (i.e. below mean sea level) the vegetation cannot survive (Morris et al., 2002; Koppel et al., 2005). Reattachment, which is much slower, may occur where the bank is not as steep and may be

facilitated by sediment trapping on the bank side of the block coupled with vegetation growth. The gaps between slump blocks and the marsh platform are able to accumulate a large amount of sediment and increase elevation as a result of reduced water velocity behind vegetated patches and the proximity to channel edges (Bouma et al., 2007; D'Alpaos et al., 2011; Marani et al., 2007; Temmerman et al., 2003b, 2003a). Field observations have confirmed this higher sedimentation within gaps, showing that local sedimentation rates in gaps are approximately 1.5-2 times higher than on the adjacent marsh

platforms (Gabet, 1998). In addition to these physical processes, the vegetative spread of *Spartina* facilitates the rapid filling of these gaps as long as local elevations can support its survival (Ge et al., 2013; Huang et al., 2008; Liu et al., 2014). Numerical modeling has also indicated that sediment accumulation can potentially facilitate the re-establishment of vegetation (Koppel et al., 2005), which in turn could lead to the reconnection of slump blocks. It would be interesting to explore these opposing processes (sediment accumulation and vegetative regrowth versus erosion and elevation loss) to

determine whether they can be used to predict slump block fate. It may be that larger blocks are likely to develop sheltered areas behind them,  which can accumulate more sediments to build up the surface (Bouma et al., 2007; Le Bouteiller and Venditti, 2015), and that there are thresholds of size, gap distance, and elevation from which slumps can recover.

### 4.3 Slump block contribution to changes in channel area

Our repeated UAV observations suggest that the studied segment of Dean Creek expanded laterally between 2020 and 2021.
The widening of internal channels is consistent with the trend over recent decades (Fig. S6 in Supporting Information) as well as previous observations in Georgia marshes (Burns et al., 2021b, 2021a). Widening in interior channels have also been observed in marshes in other areas (e.g., Chen et al., 2021; Vandenbruwaene et al., 2013; Watson et al., 2017). The fact that bank slumping accounted for 66% of the increase in channel area in this study demonstrates that slumping can be an important mechanism for channel widening, and underscores the need to understand what types of marshes produce slump

blocks and what controls their formation and fate.  Our findings demonstrate that marsh loss from slump block formation in





more quiescent, protected marsh areas, where fetch and wave disturbance are limited, can be equally important as that at the open-fetch marsh edge. The rate of vegetated marsh loss observed here was about 0.41 m/yr (0.29 m/yr due to slump blocks). This rate is on the same order as open-fetch erosion that is associated with wave attack, documented as 0.23 m/year in Horse Island Marsh, DE (Schwimmer, 2001), 0.26 m/year in San Felice marsh in the lagoon of Venice, Italy (Yang et al., 2023), 410 and 0.23-2.81 m/year in marshes in Charleston, SC (Mariotti and Fagherazzi, 2013). Moreover, Burns et al. (2021) found that marsh loss along interior channels was 16-fold greater than along the open fetch edge in a GA marsh and more than 3-fold greater in marshes in Plum Island, MA. These interior processes have received less attention than lateral erosion due to wave attack, and would enhance models of marsh evolution if they were explicitly incorporated.

We anticipate that sea level rise will likely increase slump block formation and loss, further exacerbating channel widening. 415 This is consistent with results derived from numerical modeling (D'Alpaos et al., 2010; Kirwan and Murray, 2007). Rising seas serve to increase the inundation period and depth of the marsh platform, thus potentially challenging bank stability by reducing the cohesion of sediments and increasing the pore water pressure within them. The prolonged flooding of blocks may also result in more being lost due to submergence as opposed to reconnecting, with greater inundation driving mortality of stabilizing vegetation. Additionally, intensified storms may enhance the energy of currents along banks (Leonardi et al., 420 2016), potentially leading to the production of more slump blocks.

### 4.4 Implications and future work

Although there is much work to be done to evaluate how and under what circumstances slump blocks are formed and what determines their fate, the findings presented here demonstrate that slump blocks can be dynamic features of salt marshes. As UAV and other remote sensing technology become available, it will be interesting to see where else slump blocks are found 425 and whether they share common characteristics in terms of slope, sediment composition and cohesion, and tidal energy. Being able to predict whether a block reconnects or submerges is also important for future modeling of marsh persistence in a world of accelerating SLR. Our working conceptual model is that local slope determines whether a block ends up dropping to an elevation below which the vegetation can survive. Block size and the initial gap are both also important: blocks that are larger and have a smaller gap are more likely to be able to trap sediment to support revegetation.

This work also raises multiple lines of inquiry regarding the significance of slump blocks in terms of their effects on marsh geomorphology and sediment dynamics. For example, we do not yet know the ultimate fate of submerged slump blocks, whether it results in sediment loss, nor its importance to the sediment budget of the marsh and estuarine system. We suggest that a portion of sediments is likely to be redistributed over the marsh platform (Hopkinson et al., 2018; Kirwan and Guntenspergen, 2010; Mariotti and Carr, 2014; Yang et al., 2023) and the rest exported from the system (Ganju et al., 2015). 435 Exported sediments are probably widely distributed within the estuarine system, given the energetic tidal current velocities (Blanton et al., 2003). Loss of slump blocks also results in a loss of vegetated habitat, which has implications for carbon cycling. As an initial estimate, above-ground biomass of tall *Spartina* for this site was estimated as 1004 ± 349 g/m$^2$/y (Wieski and Pennings, 2014). Considering that we documented a loss of approximately 407.91 m$^2$ of the vegetated area

through slump block submergence in one year (Fig. 6f), this amounts to a large flux of annual carbon production,
emphasizing the need for further studies in this area. New and existing technologies could be applied in new ways to create a
spatially explicit estimation of the sediment loss and carbon release via slump blocks through the integration of field
observations and active (e.g., LiDAR and SAR) and passive remote sensing data (e.g., multi- or hyper-spectral data).

Isolated slump blocks also have implications for ecosystem processes. The process of block formation represents a
disturbance to intact marsh edge habitat (Li and Pennings, 2016). Depending on its elevation and longevity in the creek, this
will affect not only the vegetation but the associated fauna and also sediment geochemistry and the microbial community, all
of which will likely vary over the lifetime of the block. The blocks also represent ephemeral habitat for fish and other
nekton and may represent a hot spot for predation on marsh invertebrates. All of these questions highlight the need for
further study of slump block processes.

**Data availability**

Data analyzed in this study are available on the GCE-LTER Data Website: https://gce-lter.marsci.uga.edu/public/app/dataset_details.asp?accession=GPH-GCED-2309

**Author contribution:**

ZY: conceptualization, formal analysis, investigation, writing – original draft, visualization, writing – reviewing and editing.
CA: conceptualization, formal analysis, investigation, visualization, writing – reviewing and editing. MA: conceptualization,
formal analysis, investigation, writing – original draft, visualization, writing – reviewing and editing.

**Competing interests:**

The authors declare that they have no conflict of interest.

**Acknowledgements**

We thank Jacob Shalack, John Williams, Adam Sapp, Matt Pierce, Claudia Vernham, and Tyler Lynn with their assistance in
acquiring and processing UAV imagery and help in the field. We appreciate the comments from Steven Pennings, John
Schalles, Jeb Byers and other members of the GCE-LTER Disturbance working group. We also thank Andrea D'Alpaos and
Chao Gao for discussions regarding slump block dynamics. This is contribution 1117 of the University of Georgia Marine
Institute.





**Financial support**

This work was supported by the Georgia Coastal Ecosystems Long Term Ecological Research project, which is supported by the National Science Foundation (OCE-1832178).

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
