# Peer review of "The dynamics of marsh-channel slump blocks: an observational study using repeated drone imagery"

_Biogeosciences, 2023_

## Referee Comment (RC1)

[referee-annotated manuscript omitted]

---

## Author Comment (AC1)

**RESPONSE TO COMMENTS ON bg-2023-180 FROM Reviewer #1**

We wish to thank the Reviewer for her/his positive assessment of this manuscript. We also want to thank the Reviewer for her/his thorough review which helped to improve the quality and clarity of our manuscript.

Please note that in this document, *italics* refer to the text of the reviewers' comments, our detailed responses are in black, the old version is in  and the planning new text of the revised version is in **bold blue**. Line numbers refer to the revised version of the manuscript.

*The manuscript "The dynamics of marsh-channel slump blocks: an observational study using repeated drone imagery" presents the evolution of slump blocks developed along Dean creek in Sapelo Island (Georgia, US) through UAV orthophoto analyses from March 2020 to March 2021. The paper is simple and concise, and although it does not "dig" into the process of slump block's formation (since no other data besides the orthophotos are included to integrate the discussion), it still gives interesting information regarding a process that, personally, I did not find frequently discussed.*
*Herein I summarize the main questions and discussion. Then I present a list of specific comments (which are on the pdf as well).*
*The paper gives some context to the process, which must be introduced since it is not too obvious. Maybe more information regarding the process of slump formation taken from other studies could be useful, although I understand that there are no many studies. The analyses seem to be thorough; however, I have a few questions:*

1. *I see in the additional information that you recorded the tidal level for each survey, that is indeed important otherwise it would be difficult to define which block can be considered as submerged and which was not. It is normal that you could not make photos perfectly at the same mean sea level, however I guess you considered the slumps submerged if they were underwater in the photo, in spite of what tidal level you had in that moment. Can you add a statement regarding this? Just to clarify what you considered underwater.*

The reviewer is correct that we considered a block submerged if it was not present in the later of two consecutive images, and that tidal levels varied in different flights. However, all the flights were carried out at lower tidal conditions, with tidal elevations ranging from -1.13 to 0.02 m above MSL (Table S1). Although this introduces some uncertainty, tides during all flights were low enough that we would have seen vegetation above the water level. Moreover, submerged blocks were never observed to re-appear in the channel in subsequent images. We therefore do not believe that differences in tide height influenced our classifications. We are going to modify the text in Lines 137-138, from "" to "**Submerged blocks were those that were not visible in the later image (Fig. 2 g, h).**"

2. *I did not see any information regarding the flights. How many GCPs did you use? Where are they located? Is that drone using an RTK system? Did you measure some random points to validate the products? Can you show the GCPs on Fig. 1? I appreciate the validation considering the error derived by subjective identification of the slumps, but it could have been improved combining it with*

*the horizontal and vertical Root Mean Square Error (since you have calculated several areas and rates). Anyway, due to the high resolution I would not expect large changes.*

The images used in this study were cropped from larger (18 ha) images that were fully described in Lynn et al. (2023). The large images were georeferenced using 12 permanent GCPs, which were distributed about 100 m apart across the area. We believe it would be misleading to provide the positions of the GCPs in Fig. 1, because our study image is a subset of much larger imagery and only 6 of the permanent GCPs would be captured. The positions of these GCPs were recorded by RTK (with horizontal accuracy of ± 2.5 cm) prior to the initial flight. They served as a reference for the georectification of all images and the georectification resulted in an average root mean square error of 5-10 cm. We would like to modify the text as follows: "**Images used in this study were cropped from larger images (Lynn et al. 2023). Briefly, these images were acquired using a DJI Matrice 210 UAV equipped with a MicaSense Altum sensor during morning low tides within 1-2 hours of solar noon. Tides at the time of the flights averaged -0.55 m MSL, with a minimum of -1.13 m MSL (Table S1). The larger images were georeferenced by 12 permanent Ground Control Points (GCPs) distributed across the entire scene, which were used to produce georeferenced images using Pix4D software. Pixel resolution was 0.05 m, and georectification resulted in an average root mean square error of 2 pixels (0.0050 $m^2$). Image processing and georeferencing is described in detail in Lynn et al. (2023).** "

3. *For what concerns sediment deposition, you say in L 389 that sedimentation in the gap between block and edge is 1.5 – 2 times higher than the adjacent marsh, but you do not mention either the average rate of deposition of the study area or the rates of deposition in the gaps. In fact, how much was sedimentation inside the gaps? Then, what are the rates in the marsh? If you have measured it, it would be interesting to add them to the manuscript.*

The difference between sedimentation rates on marsh platforms and inside the marsh-block gaps were drawn from the literature, specifically Gabet (1998), as accretion rates within our study marsh are currently unavailable. We appreciate the insightful suggestion, and are interested in obtaining such measurements in our future work. To ensure clarity regarding the origin of the data mentioned, we have revised the previous text. The original statement, "", is going to be modified to **"We do not have measurements of sedimentation at the study site, but Gabet (1998) found that local sedimentation rates in marsh-block gaps were approximately 1.5-2 times higher than on the adjacent salt marsh platform in a tidal channel in San Francisco Bay, California."**

4. *One important thing that I would recommend is to improve the discussion regarding the connection between channel widening and slump formation (e.g. L 402). You say that the slump formation processes is the main mechanism of channel widening, and indeed it seems important, but (based on what you said) it looks like the block formation is a process that is separated from the channel widening itself, like it affects the channel but at the same time is something separated from it. "The fact that bank slumping accounted for 66% of the increase in channel area in this study demonstrates that slumping can be an important mechanism for channel widening, …" What I suppose you are suggesting*

*is that the erosion of the channel is occurring as slump detachment from the banks (i.e. discontinue) rather than following a continue and homogeneous lateral erosion. However, it looks to me that slump formation is one type of erosional process of the channel itself, hence I would suggest changing the storytelling of this part of the discussion highlighting how the slump formation is the main erosional process of channel widening, rather than saying that it plays an important role on the channel widening. This observation would be different if instead you meant to say that the block themselves change the flow patterns and alter bank erosion but does not look like you went in this direction to me.*

We agree that slump block formation is one of the main mechanisms of channel erosion. We would like to edit the previous text that read "". The new statement reads **"The fact that bank slumping accounted for 66% of the increase in channelized area in this study demonstrates that slumping can be an important mechanism for channel widening, and underscores the need to understand what types of marshes produce slump blocks and what controls their formation and fate."**

5.      *Overall, the paper is good and easy to understand. It just needs some corrections and to clarify some statements.*

We wish to thank the Reviewer for her/his positive response.

6.      *L 29: The marsh is vegetated by definition, I don't think that the term "vegetated" makes sense here.*

While we appreciate this suggestion, we believe that the inclusion of the word "vegetated" is important because some marshes contain bare (i.e. unvegetated) areas.

7.      *L 36/Fig. 1: I believe that it would be better to always use the letters for references (for example this one would be the Fig. 1b).*

All inset panels of Fig. 1 will be labeled as a, b, c and the text  will be adjusted.
And the next caption reads **"Figure 1: Overview of the research area. (a) the southeastern coast of the USA and the location of Dean Creek (yellow star) on Sapelo Island; (b) an example of slump blocks (photograph by Merryl Alber); (c) positions of slump blocks digitized from each image of Dean Creek and surrounding marshes, overlaying an image from July 2020 captured using a DJI Matrice 210 UAV with a MicaSense Altum (Near Infrared, central wavelength = 840 nm) . The yellow triangles indicate the positions of point bars, and the yellow arrows indicate oyster reefs; (d) the numbered reaches along the centerline of Dean Creek, separated by points where the curvature equals zero."**

8.      *L 44: Why here you use cm/month but in the discussion you use m/year? Plus, why is this value (16 cm/year) not discussed in 4.3? You say that the value you calculated later are similar to other environments, but compared to this value the average seems higher. So the slumps you investigated creeped faster then the others, although the difference is low.*

We think there has been some confusion here, as these are two different processes. The value of 16 cm/year, which is drawn from the literature (Letzsch and Frey 1980b), represents the speed of block movement once it has broken off. The rates in Section 4.3 are the change in the channel margin due to erosion. They are two different rates that are not comparable.

9.      *L 51: How? Maybe some examples could be usefull (e.g. …).*

We would like to add an example as follows: **"For example, Frey and Basan (1978) have documented that bank slumping is one of the main indications of lateral retreat of marshes in Georgia."**

10.     *L 120/Fig. 1: The Fig. 1 shows the segments. But did you think about adding the limits between the reaches in the orthophoto? It may be useful, for example using just a line to separatet each section. Unless you believe that it might become messy.*

We would like to add lines that separate the channel into different reaches, but it made the figure quite busy, especially in the upper reaches. We therefore decided that keeping the inset as it is better protrays the channel segmentation.

11.     *L 123: missing point (al.).*

This will be corrected.

12.     *L 167: the Formation.*

This will be corrected.

13.     *L 171: Delete ","*

This will be corrected.

14.     *L 230-232: Interesting, but should go in the discussion.*

This will be moved to the discussion.

15.     *L 239/Fig. 5: I suppose you used this representation because you wanted to point out that the disappearance of the slumps was mainly driven by reconnection rather than submersion, but I think it confuses the reader because here I would expect the values of the submerged blocks.*

We wish to thank the Reviewer for her/his suggestion. Indeed, Fig. 5f shows the value of submerged blocks, because it includes changes in the area in persistent blocks and totally

submerged blocks. Due to the fact that block submergence is a gradual process (see Fig. 2e-h), if we only present the area of submerged blocks in each interval (only the A in Fig. 2g), the value would be very small. It would underestimate the loss in block area by submergence and mislead the reader to think that submergence is not an important block destiny. Therefore, we believe maintaining Fig. 5f is better than only displaying the area of submerged blocks.

16.    *L 254: I may have lost this, but I did not see the explanation of what p is in the text.*

The p-value from comparing two datasets using a one-way ANOVA is referred to in Line 244. To make this clearer, we would like to change the text to state "**The initial properties of blocks that submerged (Fig. 6a) with those that were lost through reconnection (Fig. 6b) were compared using a one-way ANOVA. These analyses showed, first, that the initial size of blocks that were lost to submergence (1.45 ± 1.63 m$^2$) was significantly smaller (p-value < 0.05) than that of those that eventually reconnected (5.49 ± 4.53 m$^2$) (Fig. 6c). Second, blocks that were lost to submergence were significantly further (p-value < 0.05) from the marsh edge (1.34 ± 1.25 m) when first observed compared to those that eventually reconnected (0.49 ± 0.21 m, see Fig. 6d).** "

17.    *L 295: This part should go in the discussion. Despite this, 233 is 66% of the increase in the channel area; you should add the total increase as well, which I guess is around 350.*

We thank the reviewer for this suggestion. We would like to include the following in the discussion (Line 400): **"When taken together, the total area of new slump blocks was 886 m$^2$ and that of reconnected blocks was 652 m$^2$, resulting in a net loss in marsh area due to slump blocks of 234 m$^2$. This represents 66% of the overall increase in channel area (about 355 m$^2$), and suggests that slump blocks play an important role in tidal creek channel widening."**

18.    *Fig. 9: It is not clear to me which are the points referred to the slumps changes and the channel changes. Can you please improve the graph and make it clearer?*

Each point on the figure shows the change in channel area observed in a particular reach (y-axis) plotted against the net change in the area of slump blocks observed in that same reach (x-axis). We have re-labeled the axes "**Net change in channel area in reach (m$^2$)**" and "**Net change in channel area by slump blocks in reach (m$_2$)**", which we hope makes this clearer.

[Figure]

Figure 9: The relationship between the net change in channel area in each reach due to slump-blocks processes (the difference between the new block area and area of blocks reconnected with the marsh platform, $\Delta \boldsymbol{Ssb}$) and the observed change in channel area ($\Delta \boldsymbol{Sc}$) in each reach (see Figure 1). The lines through zero separate areas that showed loss (negative values) and gain (positive values) of channel area. The linear regression is also included along with the 95% confidence interval (represented by the light grey area).

19.    *L 325: Were the oyster reefs on both sides of the channel?*

Yes, both sides have oyster reefs. We would like to add another arrow in Fig. 1 to make that clear and change the caption: 'yellow arrows indicate areas where oyster reefs are present'.

20.    *L 383: This statement seems a bit unprecise, because it depends on what kind of hydroperiod the plants need to survive, hence not necessarily the mean sea level. However, I guess that this is what you observed in this environment. Is this true?*

The reviewer is correct. We would like to modify the text to read: **"When the block is lower than the threshold elevation for vegetation survival, vegetation is lost (Morris et al., 2002; Koppel et al., 2005)."**

21.    *L 389: Indeed, do you know the vertical changes of this study area? How much was the rate of sedimentation that you observed?*

We do not know sedimentation rates in this specific marsh and have not collected site specific data behind the blocks. We plan to measure this in the future.

22.    *L 407: I don't clearly understand how vegetation loss can be described by a linear rate (m/year instead of m2/year).*

We translated our marsh area loss number to a linear shoreline retreat rate (in meters/year) in order to be able to compare with linear rates of erosion in the literature. We calculated the retreat of the channel edges along the entire creek and measured  an average increase in channel

width at a rate of 0.41 m/year  (– see Line 286) and compared it with the overall retreat that can be attributed to slump blocks over the same time period, again averaged over the entire channel and measured an average in channel width due to slump block creation of -0.29 m/year.

23.      *L 636: This should go before the Zhao et al., 2022.*

This will be corrected.

---

## Author Comment (AC2)

**RESPONSE TO COMMENTS ON bg-2023-180 FROM Reviewer #2**

We wish to thank the reviewer for their constructive comments on the manuscript.

Please note that in this document, *italics* refer to the text of the reviewers' comments, our detailed responses are in black, and the new text of the revised version is in **bold blue**. Line numbers refer to the revised version of the manuscript.

*The paper investigates slump blocks along marsh shorelines and their impact on marsh dynamics, particularly within Dean Creek, a tidal creek on Sapelo Island, GA, USA. Using Unmanned Aerial Vehicle (UAV) imagery spanning 11 months (March 2020–March 2021), the study tracks the spatial distribution, movement, and longevity of slump blocks. Overall, the paper demonstrates the efficacy of repeated UAV surveys in monitoring short-term geomorphological processes. Here are my comments:*

1. *An aspect deserving more detailed discussion is the influence of vegetation on slump block loss. Does vegetation have any effect on the loss?*

   We agree that the presence of vegetation affects slump blocks, and we have pointed out that the presence of vegetation may prolong the persistence of slump blocks (Lines 362-365) and that larger vegetation patches may enhance block reattachment (Lines 371-374). However, the focus of this paper is the dynamics of slump block themselves as we do not currently have the data to quantify the impact of vegetation on the process. We do agree that this is an interesting aspect for future study.

2. *The discussion needs to be strengthened regarding the effect of the short observation period and whether the conclusion can be extrapolated to a larger area. Findings of this study are specific to Dean Creek, and what is the implication for other marshes or tidal creeks?*

   We agree that this initial work was focused on our study site. However, we believe that the insights we present have broad applicability. For example, the fact that ebb-oriented point bars and oyster reefs can protect the bank from slumping, that blocks have the ability to reconnect with the marsh platform, and that slump blocks contribute to marsh erosion, are all likely to be applicable to other mesotidal marsh systems. We would like to add a sentence to highlight the importance of this analysis, which reads **"We expect that the main findings of these analyses regarding the spatial distribution of slump blocks, block dynamics and block contribution to marsh loss and creek widening will be applicable to other mesotidal marsh systems."**

3. *Specific comments: Why is the slump block important? It is unclear in the abstract.*

   We would like to change the first sentence of the abstract as follows: **"Slump blocks are widely distributed features along marsh shorelines that can disturb marsh edge habitats, and affect marsh geomorphology and sediment dynamics."**

---

## Referee Report (RR1)

The authors answered to my comments and requests. The paper is clearer compared to the previous version and I want to congratulate with the authors for the work. I haven't seen improvements under the point of view of the content, but the authors answered to my comments. All the knowledge is strictly limited to the information obtained from the aerial images, so without any integration of other data of different nature (e.g. sedimentation rates) interpretations are necessarily limited. However, this is probably the point of the study, to keep it simple. I only have these last observations/comments to point out. Here I refer to the points you named in the author response:

- **Point 3.** Ok, I appreciate that you will take a step further in sedimentation analysis in the future. However, are there any values regarding sedimentation from literature? Although you cited Gabet (1998) saying "that local sedimentation rates in marsh-block gaps were approximately 1.5-2 times higher than on the adjacent salt marsh platform", it does not help to understand the magnitude of the sedimentation of the study area. It can give an idea of the typical rates of this area and, most importantly, it gives a meaning to the sentence you added; otherwise, without any reference value, it is not comprehensible.
- **Point 6.** I beg to differ since the definition of "marsh" necessarily needs the presence of vegetation; if there is no vegetation, then it should be called tidal flat. However, in this part of the paper you are talking about the spatial changes of vegetation, hence I suggest to simply change as follow:
  L29: "Marshes are dynamic environments and the area  **colonized by vegetation** can change over time, not only as the result of progradation or retreat of the open-fetch marsh edge…"

Here I add some more comments (referred to the clean version of the paper):

L100: "Images used in this study were cropped from larger images (Lynn et al. 2023). Briefly, these images…" High repetition of the word "image". Maybe you can use other terms, maybe "orthophoto"? Also, I think it is correct to the say "**The** images used in this study…"

L304: "…although most slump blocks submerge over time,…" This is not correct, since you showed that most of the blocks reconnected while just a part submerged (652 vs 234 $m^2$). I understand that you do not want to suggest that the block submergence is not important, but I would avoid to use this sentence.

It would have been nice to improve this analysis with the use of DSMs or sedimentation rates, but as you said, it can be the next step of the study.

Kind regards.

---

## Author Response (AR2)

Jack Middelburg,
responsible editor for this special issue of Biogeosciences

Feb 10, 2024

Revision of Manuscript ID: BG-2023-180 "The dynamics of marsh-channel slump blocks: an observational study using repeated drone imagery " by Yang et al.

Dear Editor,
Enclosed please find the revised version of the above manuscript after our revision, submitted for possible publication in *Biogeosciences*.

The paper was previously assessed by two anonymous reviewers, the associate editor and the editor, who suggested the revised manuscript will eventually be publishable.

In this current revision, we have complied with all recommendations for change which are well documented in our Response Letter. All suggestions to improve the manuscript have been incorporated into the new version, as requested by you and the reviewer.

We wish to thank you and the reviewer for the thorough review work and the very useful suggestions and comments, which allowed us to significantly improve the quality of our paper.

We look forward to an acknowledgment of receipt and, hopefully, to a rapid acceptance of our paper for publication in *Biogeosciences*.

Sincerely Yours,

Zhicheng Yang,
on behalf of all of the authors

**RESPONSE TO COMMENTS ON bg-2023-180 FROM Reviewer #1**

We wish to thank the Reviewer for her/his positive assessment of this manuscript. We also want to thank the Reviewer for her/his thorough review which helped to improve the quality and clarity of our manuscript.

Please note that in this document, *italics* refer to the text of the reviewers' comments, our detailed responses are in black, the old version is in  and the new text of the revised version is in **bold blue**. Line numbers refer to the revised version of the manuscript.

*The authors answered to my comments and requests. The paper is clearer compared to the previous version and I want to congratulate with the authors for the work. I haven't seen improvements under the point of view of the content, but the authors answered to my comments. All the knowledge is strictly limited to the information obtained from the aerial images, so without any integration of other data of different nature (e.g. sedimentation rates) interpretations are necessarily limited. However, this is probably the point of the study, to keep it simple. I only have these last observations/comments to point out. Here I refer to the points you named in the author response:*

*• Point 3. Ok, I appreciate that you will take a step further in sedimentation analysis in the future. However, are there any values regarding sedimentation from literature? Although you cited Gabet (1998) saying "that local sedimentation rates in marsh-block gaps were approximately 1.5-2 times higher than on the adjacent salt marsh platform", it does not help to understand the magnitude of the sedimentation of the study area. It can give an idea of the typical rates of this area and, most importantly, it gives a meaning to the sentence you added; otherwise, without any reference value, it is not comprehensible.*

As far as we know, there is no literature focused on our study area reporting sedimentation rates within the gap between marsh platform and slump blocks. To provide insight for readers into the differences in the magnitude of the sedimentation, we have modified the text by adding the sedimentation rates observed by Gabet (1998). The new text reads (Lines 389-390) "**but Gabet (1998) found that local sedimentation rates in marsh-block gaps (about 90 mm/year) were substantially higher than on the adjacent salt marsh platform (1-55 mm/year) in a tidal channel in San Francisco Bay, California.**"

*• Point 6. I beg to differ since the definition of "marsh" necessarily needs the presence of vegetation; if there is no vegetation, then it should be called tidal flat. However, in this part of the paper you are talking about the spatial changes of vegetation, hence I suggest to simply change as follow:*
*L29: "Marshes are dynamic environments and the area of vegetated marsh colonized by vegetation can change over time, not only as the result of progradation or retreat of the open-fetch marsh edge…"*

We agree with the reviewer's comment. We have modified the text in Lines 29-31 from "" to "**Marshes are dynamic environments and their vegetated area can change over time, not only as the result of progradation or retreat of the open-fetch marsh edge.**"

*Here I add some more comments (referred to the clean version of the paper):*
*L100: "Images used in this study were cropped from larger images (Lynn et al. 2023). Briefly, these images…" High repetition of the word "image". Maybe you can use other terms, maybe "orthophoto"? Also, I think it is correct to the say "The images used in this study…"*

We agree with this comment. We have edited the text from "" to "**The images used in this study were cropped from larger orthophotos (Lynn et al. 2023). Briefly, these data were acquired using a DJI Matrice 210 UAV equipped with a MicaSense Altum sensor during morning low tides within 1-2 hours of solar noon.**"

*L304: "…although most slump blocks submerge over time,…" This is not correct, since you showed that most of the blocks reconnected while just a part submerged (652 vs 234 m2). I understand that you do not want to suggest that the block submergence is not important, but I would avoid to use this sentence.*

We agree with the reviewer. We have modified the text from "" to the next text, which reads "**Second, some slump blocks submerge over time, while some of them reconnect to the intact marsh platform.**"

*It would have been nice to improve this analysis with the use of DSMs or sedimentation rates, but as you said, it can be the next step of the study.*

Thanks to the reviewer for this constructive suggestion. We will analyze changes in slump blocks and dynamics based on DSMs and measure sedimentation rates in the future.